# Exploring the Impact of Extra Virgin Olive Oil on Maternal Immune System and Breast Milk Composition in Rats

**DOI:** 10.3390/nu16111785

**Published:** 2024-06-06

**Authors:** Sonia Zhan-Dai, Blanca Grases-Pintó, Rosa M. Lamuela-Raventós, Margarida Castell, Francisco J. Pérez-Cano, Anna Vallverdú-Queralt, Maria José Rodríguez-Lagunas

**Affiliations:** 1Department of Biochemistry and Physiology, Faculty of Pharmacy and Food Science, University of Barcelona, 08028 Barcelona, Spain; soniazhan@ub.edu (S.Z.-D.); blancagrases@ub.edu (B.G.-P.); margaridacastell@ub.edu (M.C.); mjrodriguez@ub.edu (M.J.R.-L.); 2Institute of Nutrition and Food Safety (INSA-UB), University of Barcelona, 08921 Santa Coloma de Gramenet, Spain; lamuela@ub.edu (R.M.L.-R.); avallverdu@ub.edu (A.V.-Q.); 3Department of Nutrition, Food Science and Gastronomy, Faculty of Pharmacy and Food Science, University of Barcelona, 08028 Barcelona, Spain; 4CIBER Physiopathology of Obesity and Nutrition (CIBEROBN), Institute of Health Carlos III, 28029 Madrid, Spain

**Keywords:** maternal diet, breastmilk, immunoglobulin, polyphenols

## Abstract

Maternal breast milk plays a key role in providing newborns with passive immunity and stimulating the maturation of an infant’s immune system, protecting them from many diseases. It is known that diet can influence the immune system of lactating mothers and the composition of their breast milk. The aim of this study was to establish if a supplementation during the gestation and lactation of Lewis rats with extra virgin olive oil (EVOO), due to the high proportion of antioxidant components in its composition, has an impact on the mother’s immune system and on the breast milk’s immune composition. For this, 10 mL/kg of either EVOO, refined oil (control oil) or water (REF group) were orally administered once a day to rats during gestation and lactation periods. Immunoglobulin (Ig) concentrations and gene expressions of immune molecules were quantified in several compartments of the mothers. The EVOO group showed higher IgA levels in both the breast milk and the mammary glands than the REF group. In addition, the gene expression of IgA in mammary glands was also boosted by EVOO consumption. Overall, EVOO supplementation during gestation and lactation is safe and does not negatively affect the mother’s immune system while improving breast milk immune composition by increasing the presence of IgA, which could be critical for an offspring’s immune health.

## 1. Introduction

A balanced diet is important for good health and, among other body systems [1], diet plays a key role in the maintenance of the immune system’s optimal functioning [2]. The Mediterranean diet (MD) is known as a healthy lifestyle, which is characterized by a high consumption of fruits and vegetables, whole grain cereals, legumes and seeds [2]. The usage of extra virgin olive oil (EVOO) stands out by being the main source of fat in this diet [3]. EVOO is particularly beneficial for its antioxidant and anti-inflammatory properties [4]. Its high monounsaturated fatty acid (MUFA) content, of up to 80%, has always been related to all these healthy properties; however, lately, EVOO’s benefits are also related to minor components, which make up 1–2% of the total contents, such as phenolic compounds [3]. There are a great variety of phenolic antioxidants present in EVOO such as phenyl alcohols, phenolic acids, secoiridoids, flavonoids and lignans, with the main phenolic compounds being tyrosol, hydroxytyrosol, oleuropein and oleocanthal [5]. After these EVOO pehnolics’ intake, there is an increase in their plasma concentrations and urine elimination, which seem to be due to a dose-dependent absorption [6,7,8]. Then, EVOO’s antioxidants are tissue-distributed and have been linked to health properties through many different mechanisms [9,10]. In this sense, EVOO intake has been associated with an increase in the activity of glutathione peroxidase (GPX), superoxide dismutase (SOD) or catalase (CAT), enzymes implicated in antioxidant activity [11,12,13].

In addition to their antioxidant role, these compounds are also antimicrobial [4,14,15] and anti-inflammatory [16], being that all these properties are associated with the prevention of neurodegenerative diseases, cancers and rheumatic pathologies [10,17]. Nevertheless, EVOO has always had a big impact on cardiovascular diseases [18,19] and it also seems to positively modulate the intestinal microbiota [20,21]. Although it is clear the relationship between antioxidant intake, the homeostasis of the redox systems and their modulation of immune system activities and inflammation [22], little is known about the effect of EVOO on the immune system (IS). In addition, even less studied is this relationship during pregnancy and lactation periods or its impact on breast milk (BM) composition.

BM plays a key role in growth and an infant’s immune system [23]. It participates in the protection of the infant from infectious diseases during the first months of life due to its different components such as maternal immunoglobulins (Ig), hormones, cytokines, immunocompetent cells, or a variety of antimicrobial compounds. It also contains its own microbiota and prebiotic substances such as oligosaccharides (HMOs) that stimulate the growth of beneficial bacteria in the gut [24].

Therefore, breastfeeding offers a multitude of benefits; besides shielding against infections, it promotes optimal neurodevelopment, reduces allergies, lowers the chances of obesity and diabetes, and safeguards against cardiovascular diseases and asthma, among others [25]. In humans, it is recommended to exclusively breastfeed for the first 6 months and continue until the child reaches 2 years, supplemented with complementary food [26].

Besides genetics, different factors can influence breastfeeding and breast milk composition [27], such as parity, the method of delivery, the breastfeeding technique, feeding frequency, maternal nutritional status, dietary intake, and eating habits [25,28,29]. A disrupted nutritional status affects the mother’s capacity to produce sufficient BM for the infant’s nutritional needs [25].

In this line, nutritional interventions during gestation and lactation can be a strategy to modulate the properties of BM. Previous nutritional interventions, both, at preclinical and clinical levels, have shown modifications in the lipid and immune profiles of BM [30,31,32]. Furthermore, it has been recently described that plasma EVOO metabolites reach the BM of EVOO-supplemented rats [33]. 

In order to better understand the impact of EVOO on health, and particularly on the IS in these periods, a preclinical approach has been designed. The aim of this study was to assess the impact of EVOO supplementation during pregnancy and lactation on the mother rat’s immunity and on BM composition.

## 2. Materials and Methods

### 2.1. Animals

The study was performed with 20 female and 5 male Lewis rats from Envigo (Sant Feliu de Codines, Spain) aged 8 weeks in two different cohorts. Animals were individually housed in cages and fed with a standard diet, with the AIN-93G formulation [34] and water ad libitum. The rats were randomly mated with males (2 female rats and 1 male per cage) for 48 h and housed individually again. Finally, after confirming pregnancy, the experiments were carried out with 14 pregnant rats. The animals were under controlled temperature and humidity conditions, in a 12:12 h light:dark cycle in the Diagonal Campus Experimentation Unit located at the Faculty of Pharmacy and Food Science (University of Barcelona).

The study was performed according to the criteria of the Guide for the Care and Use of Laboratory Animals. All experimental procedures were reviewed and approved by the Ethical Committee for Animal Experimentation of the University of Barcelona and the Catalan Government (CEEA-UB Ref. 240/19 and DAAM10933, respectively).

### 2.2. Experimental Design

The intervention lasted 6 weeks, 3 weeks of gestation (G0–G21) and 3 weeks of lactation (L1–L21). Rats were distributed into three experimental groups: REF dams (reference, n = 4), that received water; ROO dams (refined olive oil, n = 4) that received a refined olive oil; and EVOO dams (n = 6). Water, ROO and EVOO were administered daily at dosage or 10 mL/kg of body weight, by oral gavage during both gestation and lactation. EVOO proceeded from the Picual cultivar and was purchased from a supermarket in 2020. The phenolic profile of the EVOO was analyzed and described in a previous study [33], in which there was a total content of phenolic compounds of 862 mg/kg with a high proportion of oleuropein and oleacein.

Body weight, water and feed consumption were recorded daily. On the last day of the study, the lengths (nose-anus) of the rats were measured. The body mass index (BMI, body weight/length^2^, g/cm^2^) and the Lee index ($\sqrt[{3}]{weight}/length\times 1000)$ were calculated.

### 2.3. Sample Collection and Processing

During the study, fecal samples were collected weekly to study pH and humidity, and blood samples were collected at G0, G14 and at the end of the study (L21). Milk samples were collected at L21, initiating after 5 min of administrating oxytocin (Syntocinon 10 U.I./mL, Alfasigma S.L., Bologna, Italy) and by gently stimulating the teats from base to top as carried out in previous studies [35,36]. The milk samples were collected with sterile pipette tips and sterile eppendorfs and centrifuged (800× *g*, 10 min) to obtain lactic sera that were stored at −80 °C.

After milk sample collection, all the animals were intramuscularly anesthetized with ketamine and xylazine (90 mg/kg and 10 mg/kg, respectively). Blood was drawn via cardiac puncture, collected in tubes for hematological analysis (Spincell, Monlab Laboratories), and centrifuged to obtain plasma for Ig quantification. 

The adipose tissue, mesenteric lymph nodes (MLNs), spleen, liver, thymus, salivary gland (SGs) and mammary glands (MGs) were obtained to study the gene expression via PCR (soaked in RNA later (Ambion, Life technologies, Madrid, Spain)), and were homogenated for Ig quantification or fixed for histological study. Other organs (the right kidney, heart, brain, stomach and cecum) were extracted and weighed. The small intestine (SI) was weighted and measured, and the gut wash (GW), the cecal content (CC) and samples for histology and gene expression study were obtained. 

### 2.4. Immunoglobulin and Galectin Quantification

Concentrations of IgA and IgM in the homogenates of SGs, MGs, MLNs, CCs, and the GW were quantified using rat IgA and IgM ELISA kits (Bethyl Laboratories, Montgomery, TX, USA) [37,38]. The lower limits of detection were 1.95 ng/mL for IgA and 1.95 ng/mL for IgM. The protein content of samples was quantified using a BCA protein assay (Thermofischer Scientiffic, Walthan, MA, USA). Galectins (GALs) were quantified using GAL-1, GAL-3 and GAL-9 ELISA Kits (Elabscience Biotechnology Inc., Houston, TX, USA).

Concentrations of IgA, IgM and IgG subtypes in plasma from G0, G14 and L21 and in SGs, MLN homogenates and BM at L21were quantified by a Procarta Plex Rat Antibody Isotyping Panel (eBioscience, Frankfurt, Germany), following manufacturer’s protocol as performed in previous studies [32]. The plate was run on the Luminex MAGPIX analyzer (Luminex^®^, Austin, TX, USA) at the Scientific and Technological Centers of the University of Barcelona (CCiT-UB). 

### 2.5. Immunoglobulin Quantification

The percentage of cecal bacteria and Ig-coated bacteria (Ig-CB) was determined following our group’s previously described method [39], with minor adjustments. Specifically, only 10 µL of the homogenized cecal sample was utilized. Flow cytometry analysis was conducted using a Cytek Aurora instrument (Cytek Biosciences, Inc., Fremont, CA, USA) at the CCTi-UB facility, with acquisition parameters set to yield a maximum of 25,000 counts. Data analysis was carried out using FlowJo v.10 software. 

### 2.6. Gene Expression Study

SI and MG portions kept in RNA were later homogenized for 30 s in lysing matrix tubes using a FastPrep-24 instrument (MP Biomedicals, Illkirch, France). RNA was isolated using a RNeasy^®^ Mini Kit (Qiagen, Madrid, Spain) and its concentration and purity established by a NanoPhotometer (BioNova Scientific, Fremont, CA, USA). After cDNA conversion, the specific PCR TaqMan^®^ primers and probes used to assess gene expression with real-time PCR (ABI Prism 7900 HT, Applied Biosystems, AB, Weiterstadt, Germany) are shown in Appendix A. The relative gene expressions were normalized with the housekeeping gene Gusb (Rn00566655_m1) using the 2^−ΔΔCt^ method. Results are expressed as percentage of values of each supplemented group normalized to the mean value obtained for the REF group, which was set at 100%.

### 2.7. Statistical Analysis

All results are shown as means ± SEM. Statistics were performed by the software package of the IBM Statistical Package for the Social Sciences (SPSS 22.0, Chicago, IL, USA). Shapiro–Wilk was applied to evaluate normality. When there was a normal distribution, the parametric ANOVA test was carried out and the homogeneity of variance was evaluated by applying Levene’s test. An ANOVA test was followed by a Bonferroni post hoc test when there was an equality of variance and by Dunnet when there was not. When there was a different distribution, results were analyzed via Kruskal–Wallis test followed by the Dunn post hoc test. Differences between groups were considered statistically significant for *p* values < 0.05.

## 3. Results

### 3.1. Effect of EVOO on Body Growth

The Lewis dams’ body weight gain and daily intake of food were calculated through monitoring throughout the gestation and lactation periods. Although dams did not exhibit differences in body weight until the last days of lactation (Figure 1A), the ROO and EVOO groups showed lower daily chow intakes during lactation (Figure 1B). The daily water intake showed no difference between groups (Figure 1C), but all dams consumed significantly more water during the lactation period (~38 mL) than during the gestation period (~29 mL). Morphometric variables such as the BMI and Lee Index were also evaluated and neither of them were modified by any of the supplementations (Appendix A).

### 3.2. Effect of EVOO on Intestinal Morphology and Function

Concerning intestinal morphometry, some differences appeared after oil supplementation. In particular, an increase in the relative weight and length of the small intestine was found (Figure 2A,B). However, changes in adipose tissue were not evidenced (Appendix A).

In addition, the expression levels of genes involved in epithelial barrier function such as tight junction (TJ) proteins were evaluated, but no differences were observed (Figure 2C). Similarly, the amount of the mRNA of genes related to the immune system such as Toll-like receptors (TLR) did not exhibit any difference (Figure 2D). Some lack of effect could be due to the high variability found, specially in ZO-1 and Claudin-2 or TLR2 and TLR7.

IgA and IgM levels were analyzed in cecal content and GW. There were no differences in either of them between the study groups (Figure 2E,F). However, the proportion of bacteria bound to Ig (Ig-CB) in cecal samples was significantly higher in both the ROO and the EVOO group (Figure 2G).

The study of the water content and pH in feces revealed no changes due to ROO or EVOO supplementation (Appendix A).

### 3.3. Effect of EVOO on Mucosal and Systemic Immunity

To further understand the modulation of the mucosal immune response, the profile of Igs (IgA, IgM, IgG and its subtypes) were evaluated in different mucosal-associated lymphoid tissues at the end of the study (L21). The IgA, IgM and IgG levels did not show any differences in the SGs or MLNs (Table 1).

Regarding systemic immunity, plasma IgA and IgM concentrations at the end of the study (L21) were lower in the EVOO group than in the ROO group, with no differences in the concentration of total IgG and its isotypes (Table 1). 

Considering that IgG2b and IgG2c are associated with the Th1 immune response and IgG1 and IgG2a are related to the Th2 immune response in rats, the ratio between Th1 and Th2 immune responses was calculated in the plasma, SGs and MLNs. No significant differences were observed in any tissue, although, without statistical differences, a trend pattern of a higher Th1/Th2 ratio in the EVOO group was observed in all three compartments (especially in plasma). In fact, a study of the plasma Igs levels during pregnancy and lactation at different time points (G0, G14 and L21) was performed but there were no differences between groups at the other time points (Appendix A). 

### 3.4. Effect of EVOO on Breast Milk Immune Composition

After EVOO and ROO supplementation during gestation and lactation, the Ig profile in BM was established. Although there were no differences in the milk concentration of IgM, IgG and its isotypes, and the Th1/Th2 ratio (Figure 3B,C), a two-fold increase in the IgA concentration in BM was detected (Figure 3A). 

To deepen the study of breast milk, the content of galectins 1, 3 and 9 (GAL-1, GAL-3 and GAL-9) were quantified. However, no differences were found between the studied groups (Figure 3D,E). They were also quantified in plasma but only GAL-9 was detected and it showed no differences either. In addition, leptin and adiponectin levels in breast milk were not changed due to any of the interventions (Appendix A).

In line with the results of IgA in BM, IgA levels in the mammary glands of EVOO rats were higher than those in ROO rats (Figure 3F) and the relative gene expression of IgA in the mammary glands also showed a tendency to increase (*p* = 0.073) (Figure 3G).

In previous studies, we described that EVOO metabolites are present in the breast milk of the mothers of the current study [33]. Thus, some correlations between those EVOO metabolites and Igs levels in breast milk and mammary glands were studied. Many Igs showed a positive correlation with different metabolites, especially the IgA from both breast milk and mammary glands (Appendix A).

## 4. Discussion

EVOO, the main fat source of the Mediterranean diet, is known for its antioxidant and anti-inflammatory properties due to its high levels of MUFAs and phenolic compounds [3]. However, little is known about the effect of EVOO during pregnancy and lactation periods. In previous studies, we demonstrated the presence of EVOO metabolites in plasma and breast milk of rats that were supplemented EVOO daily during both periods [33]. Specifically, many antioxidant phenolic compounds (i.e., hydroxytyrosol and tyrosol) but also their phase II and microbial-derived metabolites were observed. These results prompted us to further study the impact of this oil on maternal immunity. In the present study, we have demonstrated that EVOO supplementation during gestation and lactation is able to strengthen the immune potential of the BM by means of increasing the IgA content, among other effects. 

In the context of the body weight time course throughout gestation and lactation, differences attributable to dietary supplementation only emerged in the last days of lactation. During this period, both the ROO and EVOO groups exhibited a reduction in weight compared to the REF group. These results, showing that rats subjected to lipid supplementation weigh less, are in line with previous research indicating that the oral administration of macronutrients, such as the oil in this study, can induce a sense of satiety [20], which in turn can affect the food intake and final body weight [40]. The animals in these groups also exhibit a reduced food intake, thus confirming this mechanism as a potential consequence of this satiety. In addition, it has been shown that phenolic antioxidants could also have an impact on the hormonal control of hunger/satiety, the increase of energy expenditure, but also on adipokines and insulin levels [41]. Thus, EVOO’s phenolic antioxidants can also play a role through these mechanisms.

Regarding the relative organ weights and lengths considered, only SI weight and length were modified, with increases in both the ROO and EVOO groups. It is of interest that this effect can be observed just with 6 wk of supplementation, as others have required longer expositions to find an impact [42]. These changes, associated with the lipid supplementation, which could be related, among others, to the oleic acid activity on the epithelial growth factor pathway [43], did not have an impact in the expression levels of genes involved in epithelial barrier function such as tight junction proteins, mucins or TLR. We could assume olive oil does not negatively affect the small intestine’s structure and function.

While we did not observe any differences in IgA or IgM levels in cecal content between the study groups, the proportion of Ig-CB was 2–3-fold higher in both the ROO and EVOO groups. The examination of Ig-CB has been a subject of debate due to its imbalance in individuals with different conditions [44]. In healthy individuals, the microbiome coated with IgA plays a stabilizing role in the gut, promoting a symbiotic relationship between the host and the microbiome. However, in an inflamed gut, this IgA-CB might worsen inflammation [45]. In our model, under healthy conditions, there was no impact in intestinal IgA levels; nevertheless, it led to an increase in the bacteria coating IgA in the cecum. Although the impact of the EVOO supplementation on the microbiota composition remains to the ascertained, this finding aligns with prior research suggesting a role in maintaining gut homeostasis [46,47]. The production of cecal IgA can be triggered by both endogenous and pathogenic bacteria, with pathogen-induced IgA being considered high-affinity and specific [47,48,49]. As a result, although the total IgA might not increase, the elevated Ig-CB could be linked to an increased affinity towards pathogenic species, which could help facilitating their elimination. The results are in line with others in which some polyphenols have also elevated the fecal IgA content [50,51].

To deepen the understanding of these effects on the immune system, we quantified the concentration of different Ig in plasma and also in different mucosa-associated tissues. Interestingly, we found an increase in the IgA in BM from rats supplemented with EVOO but not in that of rats supplemented with ROO. In human milk, the predominant Ig is a variant of the IgA, known as secretory IgA (sIgA), whose structure makes it optimal for mucosal defense, such as being able to neutralize pathogens, and also preventing excessive inflammation or damage to the tissues [52,53,54,55,56]. IgA is the most crucial class of Ig supplied by BM to the infant as it modulates and promotes the development of the infant’s immune system while it is still immature [53,57]. Therefore, it plays a key role in the intestine of the infant because its own sIgA is still in development [53]. The secretion of this Ig by the mammary glands ensures its transmission to the offspring through breast milk, providing protection against infections [58]. Therefore, it is interesting to note an increase in this Ig in milk, as well as the rise in secretory IgA in the mammary glands of rats that have been consuming EVOO compared to those of ones consuming ROO, indicating that some particular compounds only found in EVOO play this important role. This novel result could be in line with the relationship established between maternal diet diversity (including antioxidant intake) and some BM immune factor concentrations [59].

Additionally, the expression of the IgA gene in the MGs revealed a tendency towards an upregulation in the EVOO group. The Ig content of BM originates from plasma cells localized in the MGs or transferred from plasma. The tendency to upregulate IgA gene expression in the MGs suggests an enhanced local production there, rather than the filtration of IgA from plasma, as the levels of IgA in plasma remain unchanged. However, it has to be considered that some non-significant reduction in plasma IgA could also be due to its transfer to milk. Although no information is available regarding the impact of other antioxidants on IgA levels in BM, it has been described that some polyphenols could induce Tregs in vivo and consequently potentiate IgA production [60]; therefore, and taking into account the tolerogenic environment required in early life and in BM, this could be a plausible mechanism involved in this EVOO effect.

Our study has a limited sample size and despite the promising results, more studies would be needed to better understand the role of EVOO in maternal immunity and BM composition. One of the main limitations in this type of study is the low volume of milk obtained, and therefore, the number of targets should be prioritized. In addition, the dose used, when extrapolated to a human equivalent, would represent 1.5 mL/kg of EVOO consumption per day, thus, future studies with lower doses are required. Furthermore, another step will be the study of the impact of maternal EVOO supplementation on the offspring’s immunity, and also to change the focus to other systems closely related and also the microbiota development.

## 5. Conclusions

Nevertheless, we can conclude that EVOO supplementation during gestation and lactation is safe and does not negatively affect the mother’s immune system. In addition, it improves BM immune composition by increasing the presence of IgA, which could be critical for the offspring’s immune health.

## Figures and Tables

**Figure 1 nutrients-16-01785-f001:**
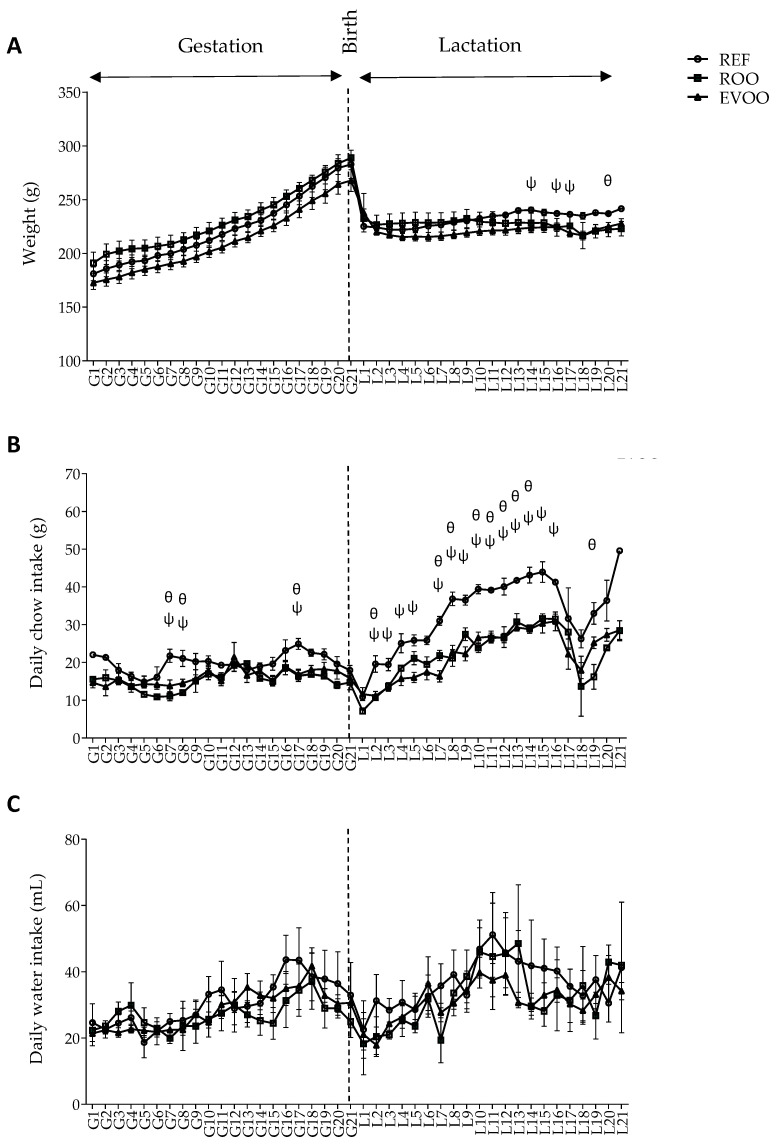
The impact of the maternal diet on growth and food and water intake along the study. The time course changes in animal weight (**A**), daily chow intake (**B**) and daily water intake (**C**). The first 3 weeks correspond to the gestation period and the following 3 weeks to the lactation period. The vertical dashed line represents the birth day. Results are expressed as mean ± SEM (n = 4–6). θ *p* < 0.05 ROO vs. EVOO. Ψ *p* < 0.05 EVOO vs. REF.

**Figure 2 nutrients-16-01785-f002:**
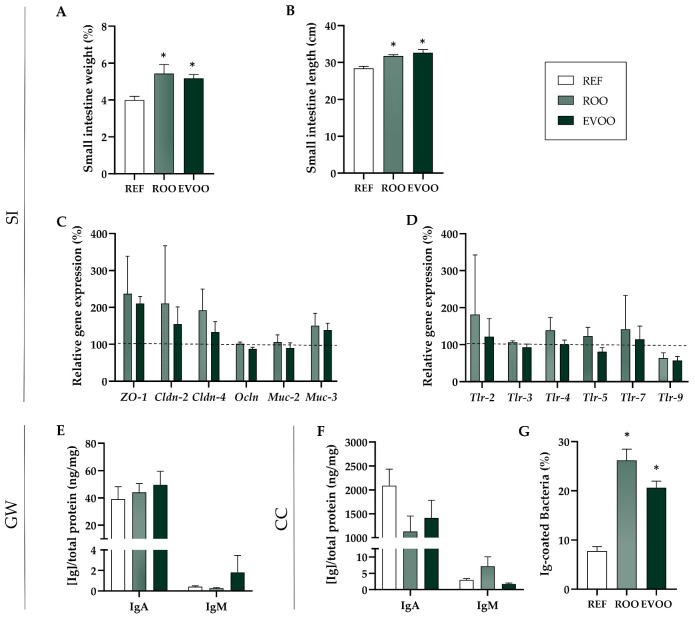
EVOO supplementation effects on intestinal morphometry and function at the end of the study (L21). Small intestine weight (**A**) and length (**B**). Organ weights expressed in g of tissue/100 g of the body. Organ length expressed in cm of tissue/100 g of the body. (**C**,**D**) The relative mRNA expression of barrier function intestinal genes (*ZO-1*, *Cldn2*, *Cldn4* and *Ocln*) and mucin genes (*Muc-2*, *Muc-3*) (**C**) and of genes related to immunity (*Tlr-2*, *Tlr-3*, *Tlr-4*, *Tlr-5*, *Tlr-7* and *Tlr-9*) (**D**) via real-time PCR, calculated with respect to REF, which corresponded to 100% of transcription (dashed line). IgA and IgM levels in gut wash (GW) (**E**). IgA and IgM levels (**F**) and Ig-coated bacteria (**G**) in cecal contents (CC). Results are expressed as mean ± S.E.M (n = 4–6). * *p* < 0.05 vs. REF.

**Figure 3 nutrients-16-01785-f003:**
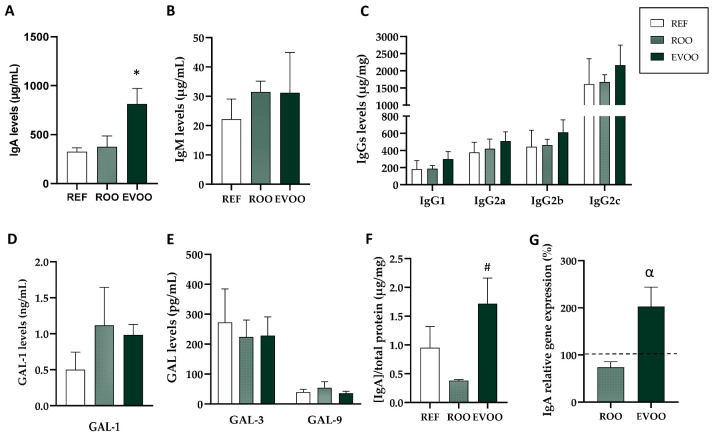
EVOO effects on breast milk. IgA (**A**), IgM (**B**) and IgG (**C**) levels in breast milk. Galectins 1 (**D**), 3 and 9 (**E**) levels in breast milk. IgA levels (**F**) and relative gene expression of IgA (**G**) in mammary gland by real-time PCR, calculated with respect to REF, which corresponded to 100% of transcription (dashed line). Results are expressed as mean ± S.E.M (n = 4–6). * *p* < 0.05 vs. REF, # *p* < 0.05 vs. ROO. α *p* < 0.07 vs. REF.

**Table 1 nutrients-16-01785-t001:** The immunoglobulin profiles after EVOO supplementation at the end of the study (L21).

**SG [Ig] (µg/mL)**	**REF**	**ROO**	**EVOO**
IgA	308.55 ± 105.79	402.61 ± 49.71	419.18 ± 55.17
IgM	13.88 ± 1.47	22.22 ± 1.94	13.72 ± 1.48
IgG	1448.74 ± 262.39	1664.06 ± 107.70	1733.56 ± 343.34
IgG1	119.83 ± 29.71	149.28 ± 23.14	122.70 ± 13.14
IgG2a	298.89 ± 52.41	324.31 ± 39.77	317.19 ± 42.47
IgG2b	257.58 ± 47.37	275.73 ± 11.02	288.08 ± 49.85
IgG2c	772.44 ± 155.16	914.74 ± 33.77	1005.58 ± 237.88
Th1/Th2	2.59 ± 0.40	2.61 ± 0.38	2.97 ± 0.47
**MLN Ig (µg/mL)**	**REF**	**ROO**	**EVOO**
IgA	62.26 ± 6.45	105.94 ± 20.19	41.88 ± 4.03
IgM	303.65 ± 46.34	522.78 ± 165.21	301.53 ± 66.49
IgG	3000.95 ± 567.51	6660.71 ± 1774.07	4697.71 ± 724.16
IgG1	83.92 ± 32.25	123.04 ± 22.64	157.36 ± 32.38
IgG2a	154.39 ± 34.20	224.49 ± 18.80	189.24 ± 48.44
IgG2b	596.48 ± 110.75	1260.58 ± 329.14	912.03 ± 132.17
IgG2c	2166.15 ± 390.31	5052.61± 1403.49	3439.07 ± 511.59
Th1/Th2	13.12 ± 2.52	17.82 ± 4.39	14.34 ± 2.20
**Plasma Ig**	**REF**	**ROO**	**EVOO**
IgA	14.20 ± 2.97	14.82 ± 0.39	10.11 ± 0.94 ^#^
IgM	77.10 ± 20.47	80.88 ± 5.52	48.27 ± 5.75 ^#^
IgG	1847.76 ± 806.02	1438.04 ± 90.52	1453.49 ± 282.67
IgG1	149.56 ± 82.09	115.25 ± 11.87	98.48 ± 15.19
IgG2a	264.04 ± 91.08	214.46 ± 33.91	210.06 ± 31.94
IgG2b	344.67 ± 148.76	254.12 ± 11.05	246.43 ± 41.12
IgG2c	1089.48 ± 484.09	854.21 ± 33.69	898.52 ± 194.42
Th1/Th2	3.58 ± 0.58	4.64 ± 1.58	5.93 ± 2.29

Results are expressed as mean ± S.E.M (n = 4–6). # *p* < 0.05 ROO vs. EVOO. Th1/Th2 = (IgG2b + IgG2c)/(IgG1 + IgG2a).

## Data Availability

The original contributions presented in the study are included in the article, further inquiries can be directed to the corresponding author.

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
