# Peer review of "Exploring the Impact of Extra Virgin Olive Oil on Maternal Immune System and Breast Milk Composition in Rats"

_nutrients, 2024, doi:10.3390/nu16111785_

Round 1

Reviewer 1 Report

Comments and Suggestions for Authors

-       Introduction is mainly concentrated on EVOO. The authors have already written about EVOO composition in their previous paper, thus, in order to highlight the knowledge gaps, more literature information about breastfeeding, breast milk and immunity also in human and nutrition would be helpful for the paper value

Does the intake of EVOO have any effect on the production, quantity and quality of breast milk?

-       Did the authors consider detecting the expression of inflammatory cytokines such as TNFα, IL1, and IL6 expression or  others, taking into account that they are modulated by EVOO polyphenols?

-       Could you specify how much t daily intake of EVOO by rats corresponds to  humans?

-       do you have any explanation for the IgA and IgG reduction in plasma?

Reviewer 2 Report

Comments and Suggestions for Authors

The proposed manuscript titled "Exploring the impact of Extra Virgin Olive Oil on maternal immune system and breast milk composition in rats" aims to reveal if a supplementation during gestation/lactation of rats with extra virgin olive oil has an impact on the mother’s immune system and on the breast milk immune composition.

Through a meticulously conducted study, the authors provided a comprehensive analysis of the impact of EVOO supplementation during pregnancy/lactation on both mother's immunity and BM composition.

The findings revealed that EVOO supplementation during gestation and lactation does not negatively affect the mother's immune system and improves breast milk immune composition.

The discussion/conclusions are well outlined and concise, clearly expressing the limitations of the proposed study.

The references reported are appropriate and include recent literature data, in line with the present study.

However, I believe that the authors should revise some points before the work is accepted for publication:

1-      The authors claim that a 6-week supplementation in rats can significantly increase both the weight and length of the small intestine. However, the exposure time to the proposed treatment seems insufficient to cause such a profound morphological alteration in the animals. Are there any literature data reporting such a phenomenon after treatment with natural compounds? The authors should provide a thorough justification for this point.

2-      The RNA expression levels reported in Figure 2 (panels C and D) show SEM values that are too high, indicating either high variability between the groups considered or insufficient sensitivity of the experimental procedure used. The authors should justify these results.
